# Minutes to Converge: Dataset Distillation for Rapid SNN Training on Event Streams

## Abstract

Event cameras generate sparse, polarity-signed streams that align with how spiking neural networks compute in time, yet image-centric dataset distillation transfers poorly to this regime. We present **PACE** (Phase-Aligned Condensation for Events), the first event-native dataset distillation framework for SNNs, *which comprises two core modules: ST-DSM and PEQ-N*. **ST-DSM** densifies spikes with residual membrane potential and aligns real and synthetic streams by matching amplitude and phase using a characteristic-function projection in feature space and a discrete Fourier transform along time. **PEQ-N** is a probabilistic quantizer whose forward pass emits hard integer frames while a straight-through estimator preserves gradients and keeps compatibility with standard event-frame pipelines. We optimize only the synthetic data with a time-expanded condensation objective on frozen teacher features, which encourages causal spatiotemporal structure and shortens convergence time. On **DVS-Gestures** with **IPC=10** at **9.29%** of the data, PACE reaches **76.5%**, about **89%** of full-data performance and **+20.4** points over a strong baseline. Similar gains appear on CIFAR10-DVS and N-MNIST and transfer across SNN backbones. PACE delivers compact, accurate surrogates that reduce storage and wall-clock time and make minutes-to-converge training practical on neuromorphic streams while opening a path to efficient on-device learning and reproducible distilled benchmarks.

## 1 Introduction

Event cameras such as the Dynamic Vision Sensor (DVS) report asynchronous brightness changes instead of conventional intensity frames, delivering microsecond level latency, extreme dynamic range, and inherently sparse spatiotemporal streams (Lichtsteiner et al., 2008; Gallego et al., 2020). In tandem with the third generation of neural computing, spiking neural networks (SNNs) provide energy proportional, event driven perception and decision making: spikes encode when information arrives, computation is naturally asynchronous, and activity is sparse by design. The pairing of DVS and SNNs has therefore become a compelling blueprint for neuromorphic vision pipelines that complement frame based systems and, in latency or power critical niches, can potentially supplant them (Rueckauer et al., 2017a; Wu et al., 2018; Ye et al., 2025).

Despite this promise, many "DVS image" datasets are created by inducing saccade-like camera motions while replaying static images or videos on a monitor (*e.g.,* the saccade-converted N-MNIST and N-Caltech101) (Orchard et al., 2015). This protocol sharpens edge contours but underrepresents native event phenomena such as fine timing structure and high-dynamic-range dynamics, reducing motion diversity and biasing the learning signal. In practice, SNNs often discretize streams into a small number of time bins to stabilize training, which increases storage, lengthens sequences, and raises optimization cost. Together, these factors hinder scalable learning on DVS data.

Dataset distillation (DD) offers a principled path: synthesize compact, information-dense surrogates that train models to near parity with the full corpus while sharply reducing storage and compute. Early work posed DD as *gradient matching* between models trained on real and synthetic data (Zhao et al., 2021). Subsequent methods matched *training trajectories* to better mimic optimization dynamics (Cazenavette et al., 2022). Recent lines pursue *distribution matching* and kernel-based surrogates to scale or obtain partial guarantees in simplified settings (Zhao et al., 2023; Nguyen et al., 2021). Yet DD for *event streams* remains largely unexplored: neuromorphic signals are sparse,

polarity-signed, and strictly time-causal, and SNN learning introduces non-differentiable spikes with cross-scale temporal dependencies, which together blunt direct transfers of image-domain DD recipes (Gallego et al., 2020).

In this paper, we propose *PACE* (Phase-Aligned Condensation for Events), an event-native dataset distillation framework aligned with what SNNs compute over time. It comprises two core modules, **ST-DSM** and **PEQ-N**. **ST-DSM** densifies spikes with residual membrane potential and then matches real and synthetic streams by projecting features with random characteristic-function directions and applying a temporal FFT so amplitude and phase statistics align. **PEQ-N** is a probabilistic event quantizer whose forward pass emits hard integer frames and whose backward pass uses a straight-through estimator to keep gradients while remaining compatible with standard event-frame pipelines. We couple these pieces with a time-expanded condensation loss on frozen teacher features and optimize only the synthetic data, which encourages causal spatiotemporal structure and shortens convergence time. Extensive experiments demonstrate the effectiveness of our PACE across both binary and integer grids on several widely used event datasets. On **DVS-Gesture** with **IPC=10** at a ratio of $9.29\%$, PACE reaches $76.5\%$, about $89\%$ of full-data performance and $+20.4$ points over NCFM under the same setting.

**Contributions.** (1) We present the first dataset distillation framework for SNNs on event streams and establish a standardized benchmark, introducing the event-native method **PACE**. (2) We make temporal phase observable by densifying spikes with residual membrane potential, and we align real and synthetic streams via CF in feature space and FFT along time within **ST-DSM**. (3) We maintain pipeline compatibility while preserving gradients by introducing **PEQ-N**, a straight-through probabilistic event quantizer that outputs hard integer frames, and a time-expanded condensation objective on frozen teacher features that updates only the synthetic data. (4) We demonstrate large reductions in storage and training time while maintaining high accuracy across SNN backbones and datasets, enabling minutes-to-converge training under tight memory and latency budgets.

## 2 RELATED WORK

**Dataset Distillation** aims to compress a large training set into a compact synthetic set that preserves downstream accuracy (Wang et al., 2018). Early approaches align point-wise representations from a feature extractor via simple Euclidean losses, minimizing $\|f(x) - f(\tilde{x})\|_2^2$ between real and synthetic features (Wang et al., 2022; Zhou et al., 2022). A second line matches distributional statistics using Maximum Mean Discrepancy (MMD), which aligns higher-order moments in an RKHS and has been instantiated for dataset condensation (Zhang et al., 2024; Zhao & Bilen, 2022; Zhao et al., 2023). Parallel to these, gradient-matching methods seek to align the gradients induced by real and synthetic batches on the same network (Zhao et al., 2021); while effective, they are slow and memory-intensive, and backpropagation-through-time further exacerbates the cost in SNNs. To balance accuracy with wall-clock/compute in standard ANNs, distribution-matching (DM) variants replace gradient alignment with feature-distribution alignment, often yielding faster and more stable optimization (Zhao & Bilen, 2022). However, many feature-matching and MMD objectives predominantly constrain *magnitudes* of statistics and overlook *phase* information across channels and time. *Concurrently*, our Neural Characteristic Function Matching (NCFM) reframes distribution matching as a min–max game and matches both amplitude and phase through a neural characteristic function, providing a principled alternative to MSE/MMD surrogates (Wang, 2025).

**Spiking Neural Networks (SNNs)** compute with discrete spikes and leverage temporal coding, yielding energy and latency gains on neuromorphic hardware (Merolla et al., 2014). High accuracy is achieved either by *ANN→SNN conversion* (Rueckauer et al., 2017b; Deng & Gu, 2021; Bu et al., 2022), which calibrates activations and simulation length to cut conversion error and time steps, or by *direct training* with surrogate gradients, where STBP (Wu et al., 2018) enables deep training and later refinements tune surrogate shapes (Li et al., 2021; Wang et al., 2023), add time-aware normalization (tdBN (Zheng et al., 2021)), and encourage few-step inference (TET (Deng et al., 2022)). SNN inputs are either static frames unfolded over $T$ steps or event streams binned into temporal grids (Lichtsteiner et al., 2008; Amir et al., 2017); this unrolling makes training cost scale with $T$ under BPTT and surrogate gradients, motivating *dataset distillation* to reduce expensive updates while preserving accuracy. Crucially, event data are sparse and polarity-asymmetric and use integer or binary grids, and SNNs carry internal state (*e.g.,* membrane potentials and thresholds),

so RGB-oriented distillation objectives built around continuous intensities and CNN features may transfer poorly. While dataset distillation is well studied on frame-based benchmarks, there is, to our knowledge, no systematic study on *event datasets* for SNNs, which motivates our focus.

**Motivation and Positioning of Our Work.** Event streams encode information primarily in time, yet most dataset distillation methods are image-centric and rely on dense tensors or coarse binning, which mismatches sparse, polarity-signed, causal spikes and non-differentiable firing. This gap inflates storage and compute, complicates optimization, and weakens transfer to SNNs. We advocate an event-native, SNN-centric distillation paradigm that respects temporal structure with phase-aware alignment while remaining compatible with standard event-frame pipelines, enabling rapid SNN training under tight memory and latency budgets.

## 3 PRELIMINARY

**Neuron model.** We briefly review the widely-used neuron model, *i.e.,* the Leaky Integrate-and-Fire (LIF) (Gerstner et al., 2014). The membrane potential and spike firing of LIF model are given by:

$$H[t] = V[t-1] + \tau(X[t] - (V[t-1] - V_{\text{reset}})), \ S[t] = \Theta(H[t] - V_{\text{th}}), \ V[t] = H[t](1 - S[t]) + V_{\text{reset}}S[t], \quad (1)$$

where $\tau$ is the leaky factor ($\tau > 1$), $X[t]$ and $V[t]$ denote the input and membrane potential remaining at time step $t$. A binary spike $S[t]$ is emitted when the membrane potential $H[t]$ exceeds the threshold $V_{\text{th}}$. This is determined by the Heaviside function $\Theta(v)$, which outputs 1 if $v \geq 0$ and 0 otherwise. After firing, the potential resets to $V_{\text{reset}}$. Otherwise, it remains at $H[t]$.

**Event data.** Event cameras (Dynamic Vision Sensors, DVS) offer microsecond-level temporal resolution, low latency, and a high dynamic range ($>120\,\text{dB}$), making them highly compatible with spiking neural networks (SNNs). Concretely, they asynchronously emit a positive or negative event at a pixel whenever the brightness change exceeds a threshold, forming a sparse event stream. Each event can be written as a 4-tuple

$$\boldsymbol{e}_i = (t_i, x_i, y_i, p_i), \quad (2)$$

where $t_i$ is the timestamp, $(x_i, y_i)$ are the spatial coordinates, and $p_i$ is the polarity (1/0 or $1/-1$); $i$ indexes the $i$-th event in the stream. Although recent works explore alternative event representations (Zhu et al., 2019; Lagorce et al., 2016; Sironi et al., 2018) to better exploit DVS characteristics, the strongest-performing pipelines to date still aggregate events into frames over fixed time windows.

In our pipeline, we discretize time into $T$ bins and pack the raw stream into an event tensor $\boldsymbol{E} \in \mathbb{R}^{T \times C \times H \times W}$ (with $C \in \{1, 2\}$ depending on whether polarities are merged or separated). With a finite bin width, this representation is inherently *integer-driven* (per-bin counts); only in the theoretical asynchronous limit $T \to \infty$ (infinitesimal bins) does it converge to the native *0–1 event-driven* process. To probe purely event-driven behavior and the theoretical case of distilling directly from streams, in our `bin` setting we disable within-bin accumulation and keep at most one spike per pixel per bin (0/1 occupancy).

**Dataset distillation via distribution matching.** Because gradient-matching DC is slow and memory-hungry (and even more so for SNNs), we adopt distribution matching (DM) for efficiency. A classical DM objective aligns feature distributions produced by a fixed (or EMA-snapshotted) encoder $f(\cdot)$ under random augmentations $a \sim \mathcal{A}$:

$$\mathcal{L}_{\text{DM}}(\tilde{\mathcal{X}}) = \mathbb{E}_{a \sim \mathcal{A}} \left\| \underbrace{\frac{1}{|B|} \sum_{x \in B} \phi(f(a(x)))}_{\text{real features}} - \underbrace{\frac{1}{|\tilde{B}|} \sum_{\tilde{x} \in \tilde{B}} \phi(f(a(\tilde{x})))}_{\text{synthetic features}} \right\|_2^2, \quad (3)$$

where $B$ and $\tilde{B}$ are real and synthetic minibatches and $\phi$ selects the feature type (intermediate activations, logits, or BN statistics) (Zhao & Bilen, 2022). Yet magnitude-only statistics can miss phase-sensitive structure. Neural Characteristic Function Matching (NCFM) (Wang et al., 2025) matches the empirical characteristic functions (CFs) of features $z = f(x)$ over a frequency set $\Omega$:

$$\hat{\varphi}_z(\omega) = \frac{1}{B} \sum_{b=1}^{B} e^{i \omega^\top z_b}, \quad \mathcal{L}_{\text{CF}} = \sum_{\omega \in \Omega} \|\hat{\varphi}_{z(\text{syn})}(\omega) - \hat{\varphi}_{z(\text{real})}(\omega)\|_2^2, \quad (4)$$

which constrains both magnitude and phase, capturing finer structure than magnitude-only matching.

Figure 1: The framework of our PACE (Phase-Aligned Condensation for Events). Float synthetic data is quantized into integer-driven events via our PEQ-N module. Then, both real and synthetic event streams are fed into unified SNN teacher and student models. The core of our approach is the Spatial-Temporal Densified Spike Matching module, which trains the synthetic data by forcing its resulting spike patterns to closely mimic those generated by the real data across both space and time.

## 4 METHOD

We present *PACE* (Phase-Aligned Condensation for Events), a plug-and-play SNN dataset distillation framework that learns synthetic event data initialized from random noise. Sec. 4.1 introduces **ST-DSM**, which densifies sparse spikes via residual-membrane-potential injection and aligns amplitude/phase using the Characteristic Function (CF) in the spatial domain and the Fast Fourier Transform (FFT) in the temporal domain. Sec. 4.2 presents **PEQ-N**, which outputs hard integer event frames in the forward pass and employs a straight-through estimator for gradients, while remaining compatible with standard event-frame pipelines. The overall framework of PACE is given in Fig. 1.

### 4.1 SPATIAL-TEMPORAL DENSIFIED SPIKE MATCHING (ST-DSM)

**Spike features should be densified for phase-aware matching.** In ANNs, intermediate features are continuous (*e.g.,* ReLU activations), so CF-based matching can directly compare amplitudes and phases in feature space. In SNNs, however, layer-wise features are binarized spikes $s_t^{(l)} \in \{0,1\}^D$ and are extremely sparse in space *and* time, which makes frequency-domain phase estimation brittle. To make this contrast explicit, let $a_t \in \mathbb{R}^D$ denote a continuous ANN feature and $s_t \in \{0,1\}^D$ a SNN feature. Their (per-time) characteristic functions (CFs) under a random projection $\omega \sim \mathcal{N}(0, I_D)$ are:

$$\Phi_a(\omega, t) = \mathbb{E}\left[e^{\,\mathrm{i}\,\omega^\top a_t}\right], \qquad \Phi_s(\omega, t) = \mathbb{E}\left[e^{\,\mathrm{i}\,\omega^\top s_t}\right]. \tag{5}$$

Then we approximate $s_t$ by a Bernoulli vector with spike rate $\rho_t$ aligned with the all-ones direction:

$$\Phi_s(\omega, t) \approx (1 - \rho_t) + \rho_t e^{\,\mathrm{i}\,\omega^\top \mathbf{1}} \xrightarrow{\text{phase}} \arg \Phi_s(\omega, t) = \arg\big((1 - \rho_t) + \rho_t e^{\mathrm{i}\theta}\big), \text{ where } \theta = \omega^\top \mathbf{1}. \tag{6}$$

When $\rho_t$ is small and spikes arrive irregularly, the phase $\arg \Phi_s(\omega, t)$ exhibits abrupt jumps tied to rare spike events, offering poor temporal-phase observability. By contrast, for continuous $a_t$, $\nabla_{a_t} \Phi_a(\omega, t) = \mathrm{i}\,\omega\, \Phi_a(\omega, t)$, so $\Phi_a(\omega, t)$ varies smoothly with $a_t$. Consequently, small drifts in $a_t$ induce a smoothly varying phase, which is the behavior exploited by CF-based phase matching.

**Densified spike representation (DSR) via residual membrane potential.** To endow SNN features with a smooth temporal carrier while *retaining explicit spike decisions*, we adopt the LIF notation in Eq. 1: for layer $l$, the pre-reset membrane potential is $H^{(l)}[t]$, the spike is $S^{(l)}[t] = \Theta\big(H^{(l)}[t] - V_{\mathrm{th}}^{(l)}\big)$, and $V_{\mathrm{th}}^{(l)}$ is the threshold. We inject the *residual (subthreshold) membrane potential* into the spike stream to obtain a *single, continuous, spatio-temporally densified* feature per site:

$$\tilde{S}^{(l)}[t] = S^{(l)}[t] + \big(1 - S^{(l)}[t]\big) \frac{H^{(l)}[t]}{V_{\mathrm{th}}^{(l)}} \in (-\infty, 1]. \tag{7}$$

By Eq. 1, when a spike occurs ($S^{(l)}[t] = 1$), the potential resets. Otherwise ($S^{(l)}[t] = 0$), it remains $H^{(l)}[t]$. Thus $\tilde{S}^{(l)}[t] = 1$ at spike times and equals the normalized subthreshold value between spikes. Consequently, the CF of $\tilde{S}[t]$ admits a non-trivial subthreshold gradient even when no spike is fired:

$$\nabla_{H[t]} \mathbb{E}\left[e^{\,\mathrm{i}\,\omega^\top \tilde{S}[t]}\right] = \frac{\mathrm{i}}{V_{\mathrm{th}}} \mathbb{E}\left[e^{\,\mathrm{i}\,\omega^\top \tilde{S}[t]}\, \omega \odot \big(1 - S[t]\big)\right]. \tag{8}$$

We omit the layer superscript $(l)$. The mask $(1 - S[t])$, from $S[t] = \Theta(H[t] - V_{\mathrm{th}})$ in Eq. 1, restores phase sensitivity on subthreshold steps, which stabilizes temporal alignment for CF-based matching.

**CF in the spatial domain and FFT in the temporal domain for phase-aware matching.** Event streams exhibit rich temporal dynamics, so purely spatial matching is insufficient. Leveraging the LIF dynamics in Eq. 1, we propose **spatial-temporal densified spike matching (ST-DSM)**: combine CF-based projections in feature space with a discrete FFT along time to capture and align temporal phase. Formally, let $X, Y \in \mathbb{R}^{B \times T \times D}$ be densified features (Eq. 7) from real and synthetic batches at a chosen layer. We sample $M$ random directions $\omega_m \sim \mathcal{N}(0, I_D)$ and compute empirical CFs at each time step ($t \in \{0, 1, \dots, T-1\}$, $m \in \{0, 1, \dots, M-1\}$):

$$Z_r(m, t) = \frac{1}{B} \sum_{b=0}^{B-1} \exp\big(\mathrm{i}\,\omega_m^\top X_{b,t,:}\big), \; Z_s(m, t) = \frac{1}{B} \sum_{b=0}^{B-1} \exp\big(\mathrm{i}\,\omega_m^\top Y_{b,t,:}\big). \tag{9}$$

We then apply a full discrete Fourier transform along time (normalized with `forward`) to expose *temporal phase*:

$$F_r(m, \nu) \;=\; \frac{1}{T} \sum_{t=0}^{T-1} Z_r(m, t)\, e^{-\mathrm{i}2\pi\nu t/T}, \; F_s(m, \nu) \;=\; \frac{1}{T} \sum_{t=0}^{T-1} Z_s(m, t)\, e^{-\mathrm{i}2\pi\nu t/T}, \tag{10}$$

where $\nu \in \mathcal{I}_T = \{0, 1, \dots, T-1\}$ indexes temporal frequencies. Let $A_r = |F_r|$, $A_s = |F_s|$, and $\Delta\Phi(m, \nu) = \arg F_r(m, \nu) - \arg F_s(m, \nu)$. The ST-DSM loss at the chosen layer is

$$\mathcal{L}_{\text{ST-DSM}}^{(l)} = \frac{1}{TM} \sum_{m=0}^{M-1} \sum_{\nu \in \mathcal{I}_T} \Big[\alpha\big(A_r(m, \nu) - A_s(m, \nu)\big)^2 + 2\beta A_r(m, \nu) A_s(m, \nu)\big(1 - \cos\Delta\Phi(m, \nu)\big)\Big]^{\frac{1}{2}}, \tag{11}$$

where $\alpha, \beta \in [0, 1]$ weight amplitude and phase terms. Because $\tilde{S}_t$ (Eq. 7) mixes explicit spikes with a dense subthreshold trajectory, the temporal FFT (Eq. 10) becomes sensitive to phase shifts that would be invisible on sparse $\{0, 1\}$ spikes alone.

**Overall condensation loss.** We combine ST-DSM with a time-expanded cross-entropy loss $\mathcal{L}_{\text{CE}}$. Let $\{\mathbf{z}_t \in \mathbb{R}^C\}_{t=0}^{T-1}$ be the logits over time for a synthetic sequence. The condensation objective is

$$\mathcal{L}_{\text{condense}} \;=\; \lambda_{\text{in}}\, \mathcal{L}_{\text{ST-DSM}}^{(l)} \;+\; \lambda_{\text{inter}} \mathcal{L}_{\text{CE}}\big(\frac{1}{T} \sum_{t=0}^{T-1} \mathbf{z}_t, y\big), \tag{12}$$

where $l$ is set to be the last layer of feature extractor (before the linear layer). This objective is minimized only w.r.t. the synthetic data, and the teacher remains frozen during distillation.

## 4.2 PEQ-$N$: PROBABILISTIC EVENT QUANTIZER

**Why DVS dataset distillation needs an event quantizer?** Event streams are commonly converted into *integer-valued event frames* by temporally binning raw binary events within short windows. This practice (i) aligns with standard training/evaluation pipelines and hardware interfaces that expect frame-like tensors; (ii) removes the arbitrary ordering of events inside a bin while preserving per-pixel counts; (iii) yields a bounded, discrete support $\{0, \dots, N-1\}$ that matches categorical modeling and stabilizes optimization; and (iv) keeps compatibility with inference-time representations used by off-the-shelf SNN backbones. To faithfully mirror this evaluation protocol, we impose the **same integer constraint** on our synthetic dataset.

**Discrete events with gradients.** To keep integer event frames for inference *and* retain gradients during distillation, we employ a probabilistic event quantizer, *PEQ-$N$*. At each spatio-temporal location $r = (t, h, w, c)$, PEQ-$N$ predicts an $N$-way categorical distribution (temperature $\tau > 0$) whose forward pass produces *hard* integers, while a straight-through estimator (STE) carries gradients so that the losses in Eq. 12 back-propagate to the data parameters.

**Formulation.** Let $\mathbf{z}_r = (z_{r,0}, \dots, z_{r,N-1})$ be logits for location $r$. We define

$$p_{r,n} \;=\; \frac{\exp(z_{r,n}/\tau)}{\sum_{j=0}^{N-1} \exp(z_{r,j}/\tau)}, \qquad y_r^{\text{soft}} \;=\; \sum_{n=0}^{N-1} n\, p_{r,n}, \qquad y_r^{\text{hard}} \;=\; \arg\max_n p_{r,n}. \tag{13}$$

We combine discrete forward and continuous backward via STE:

$$y_r \;=\; y_r^{\text{hard}} \;+\; \big(y_r^{\text{soft}} - \text{stopgrad}(y_r^{\text{soft}})\big). \tag{14}$$

The gradients would flow through $y^{\text{soft}}$. Using the softmax derivative, we have:

$$\frac{\partial p_{r,n}}{\partial z_{r,j}} \;=\; \frac{1}{\tau}\, p_{r,n}\big(\delta_{nj} - p_{r,j}\big), \tag{15}$$

the analytic gradient of the soft expectation is given by:

$$\frac{\partial y_r^{\text{soft}}}{\partial z_{r,j}} \;=\; \sum_{n=0}^{N-1} n \cdot \frac{\partial p_{r,n}}{\partial z_{r,j}} \;=\; \frac{1}{\tau}\, p_{r,j}\left(j - \sum_{n=0}^{N-1} n\, p_{r,n}\right) \;=\; \frac{1}{\tau}\, p_{r,j}\big(j - \mathbb{E}_p[n]\big). \tag{16}$$

Because $|j - \mathbb{E}_p[n]| \le N - 1$ and $0 \le p_{r,j} \le 1$, the gradient is bounded:

$$\left|\frac{\partial y_r^{\text{soft}}}{\partial z_{r,j}}\right| \;\le\; \frac{N-1}{\tau}. \tag{17}$$

For the binary special case $N = 2$, $y_r^{\text{soft}} = p_{r,1}$ and $\partial y_r^{\text{soft}}/\partial z_{r,1} = (1/\tau)\, p_{r,1}(1 - p_{r,1})$ with maximum $1/(4\tau)$. Compared with direct `round` or hard thresholding (almost everywhere zero gradient and undefined at jumps), Eq. 16 and Eq. 17 provide a smooth, bounded gradient channel; the temperature $\tau$ acts as an annealing knob for the smoothness-sharpness trade-off.

**Placement and coupling.** PEQ-$N$ operates at the output of the synthetic data parameters: $\hat{\mathbf{Y}} \xrightarrow{\text{PEQ--}N} \hat{\mathbf{Y}}^{\text{hard}} \in \{0, \dots, N-1\}^{T \times H \times W \times C}$. The hard integers feed the frozen teacher to extract densified features $\tilde{S}$ (Eq. 7) for the inner ST-DSM objective (Eq. 11) and to produce time-varying logits for the discriminative term in the condensation loss (Eq. 12). During backpropagation, gradients propagate through $y^{\text{soft}}$ (Eq. 14) into both the quantizer and the synthetic data parameters.

## 5 EXPERIMENTS

**Implementation details.** Experiments are conducted on NVIDIA A40 GPUs using the BrainCog platform (Zeng et al., 2023). For a fair comparison, We adopt the VGGSNN (Deng et al., 2022) backbone across all experiments. For evaluation, we replicate existing coreset selection methods (*Random*, *Herding*(Welling, 2009), *K-Center*(Gonzalez, 1985)) and dataset distillation baselines (*DC* (Zhao et al., 2021), *DM* (Zhao & Bilen, 2022), *NCFM* (Wang et al., 2025)) under spiking settings, and validate our proposed method in the same framework. Our method is designed to be plug-and-play and is integrated into the state-of-the-art NCFM pipeline. To improve efficiency, we significantly reduce the number of distillation iterations: for NCFM, we use only 5,000 iterations (1/4 of the original 20,000). All datasets are resized to an input resolution of $48 \times 48$. For CIFAR10-DVS and N-MNIST, we distill synthetic datasets with Images-Per-Class (IPC) values of 1, 10, and 50. For the smaller DVS-Gesture dataset, we adopt IPC values of 1, 5, and 10. The ratio of synthetic to full dataset size is reported in Table 1. For the $M$ directions, we set it to 64 for all methods.

**Distillation strategies for different data types.** We apply two tailored distillation strategies based on the input format: **(1) Binary event data:** When the raw data consists of binary spikes, we also generate binary synthetic samples. We set the discretization parameter $N$ to 2 and use a high learning rate of 1.0 to encourage convergence to binary-like values (0 or 1). **(2) Integer event data:** When the data contains integer spike counts, we set $N$ to 8 to allow richer quantization. A smaller learning rate (*e.g.,* $10^{-2}$) is used to facilitate finer-grained optimization of discrete-valued outputs.

**Evaluation protocol.** We follow a rigorous evaluation setup to ensure stability and reproducibility. For each IPC setting, we run dataset distillation using 5 random seeds to produce 5 independent synthetic datasets. Each synthetic set is then used to train and evaluate 10 independently initialized models. We report the mean and standard deviation over the resulting 50 trials ($5 \times 10$), covering both training variance and synthetic data diversity.

### 5.1 MAIN RESULTS

**ANN-based DD methods fail to transfer.** As shown in Table 1, distillation methods originally developed for RGB/frame-based ANNs, such as DC and the recent NCFM, do not maintain their advantage when directly applied to event-based SNN-DVS settings. For instance, on DVS-Gesture with `int` data at IPC=10, **NCFM** achieves only $56.1\%$, which is *lower than all coreset methods*, including K-Center ($61.2\%$), Herding ($61.7\%$), and Random ($62.1\%$). Similar trends hold across

Table 1: Dataset distillation results (Top-1 accuracy, %). For all DD methods, synthetic sets are *initialized from random noise* and optimized with a single VGGSNN backbone, which is also used to train and evaluate on the distilled data. Coreset Selection baselines operate on real samples. *bin/int* denote binary and integer event grids. IPC is images per class. Ratio is the fraction of synthetic data relative to the full training set. "Full Dataset" trains the same VGGSNN on the entire real set. Results are reported as mean $\pm$ std across runs.

| Dataset | Type | IPC | Ratio/% | Coreset Selection | | | Dataset Distillation (DD) | | | | Full Dataset |
| --- | --- | --- | --- | --- | --- | --- | --- | --- | --- | --- | --- |
| | | | | Random | Herding | K-Center | DC | DM | NCFM | NCFM+PACE | |
| CIFAR10-DVS | bin | 1 | 0.1 | $16.2_{\pm0.2}$ | $18.9_{\pm0.3}$ | $10.3_{\pm0.3}$ | $19.9_{\pm0.6}$ | $15.0_{\pm0.9}$ | $23.1_{\pm0.1}$ | $\mathbf{27.3_{\pm1.2}}$ | |
| | | 10 | 1 | $23.8_{\pm0.1}$ | $24.2_{\pm0.5}$ | $19.2_{\pm0.5}$ | $21.2_{\pm1.3}$ | $20.0_{\pm0.8}$ | $22.3_{\pm0.7}$ | $\mathbf{31.3_{\pm1.4}}$ | $53.4_{\pm0.5}$ |
| | | 50 | 5 | $34.7_{\pm0.4}$ | $32.0_{\pm0.3}$ | $30.1_{\pm0.3}$ | $19.8_{\pm1.6}$ | $20.6_{\pm1.0}$ | $24.6_{\pm0.6}$ | $\mathbf{41.9_{\pm0.5}}$ | |
| | int | 1 | 0.1 | $15.4_{\pm0.1}$ | $18.3_{\pm0.1}$ | $9.9_{\pm0.3}$ | $24.6_{\pm0.9}$ | $16.3_{\pm0.8}$ | $27.4_{\pm1.0}$ | $\mathbf{34.1_{\pm2.4}}$ | |
| | | 10 | 1 | $24.1_{\pm0.2}$ | $25.4_{\pm0.3}$ | $19.9_{\pm0.5}$ | $32.2_{\pm1.3}$ | $25.0_{\pm0.9}$ | $28.9_{\pm3.1}$ | $\mathbf{41.3_{\pm1.9}}$ | $62.9_{\pm0.2}$ |
| | | 50 | 5 | $36.8_{\pm0.4}$ | $34.8_{\pm0.7}$ | $34.5_{\pm0.2}$ | $31.1_{\pm1.5}$ | $22.8_{\pm0.9}$ | $34.0_{\pm0.5}$ | $\mathbf{49.7_{\pm1.2}}$ | |
| N-MNIST | bin | 1 | 0.017 | $43.4_{\pm1.4}$ | $48.4_{\pm1.7}$ | $18.9_{\pm0.7}$ | $59.1_{\pm4.0}$ | $37.3_{\pm3.4}$ | $67.1_{\pm4.0}$ | $\mathbf{84.4_{\pm1.6}}$ | |
| | | 10 | 0.17 | $84.7_{\pm0.2}$ | $79.8_{\pm0.6}$ | $82.6_{\pm1.0}$ | $62.4_{\pm15.2}$ | $46.5_{\pm5.0}$ | $87.2_{\pm4.0}$ | $\mathbf{91.8_{\pm0.8}}$ | $99.0_{\pm0.1}$ |
| | | 50 | 0.83 | $92.9_{\pm0.1}$ | $90.1_{\pm0.6}$ | $90.9_{\pm0.4}$ | $47.6_{\pm14.3}$ | $54.7_{\pm8.2}$ | $92.4_{\pm0.0}$ | $\mathbf{94.1_{\pm0.1}}$ | |
| | int | 1 | 0.017 | $58.4_{\pm0.6}$ | $63.6_{\pm0.2}$ | $18.7_{\pm0.4}$ | $63.1_{\pm5.5}$ | $47.8_{\pm6.4}$ | $71.8_{\pm2.1}$ | $\mathbf{84.7_{\pm0.5}}$ | |
| | | 10 | 0.17 | $85.8_{\pm0.4}$ | $80.3_{\pm0.4}$ | $85.5_{\pm0.5}$ | $85.4_{\pm0.5}$ | $44.0_{\pm3.5}$ | $86.6_{\pm1.0}$ | $\mathbf{90.0_{\pm1.2}}$ | $99.3_{\pm0.0}$ |
| | | 50 | 0.83 | $93.9_{\pm0.1}$ | $89.8_{\pm0.3}$ | $91.3_{\pm0.2}$ | $91.1_{\pm1.7}$ | $46.5_{\pm7.5}$ | $89.8_{\pm0.6}$ | $\mathbf{94.8_{\pm0.1}}$ | |
| DVS-Gesture | bin | 1 | 0.93 | $28.0_{\pm1.9}$ | $40.0_{\pm0.6}$ | $14.1_{\pm1.1}$ | $40.4_{\pm4.9}$ | $21.8_{\pm2.3}$ | $33.6_{\pm1.3}$ | $\mathbf{50.7_{\pm2.1}}$ | |
| | | 5 | 4.64 | $53.9_{\pm1.2}$ | $57.3_{\pm0.6}$ | $49.2_{\pm0.5}$ | $35.6_{\pm4.9}$ | $48.1_{\pm3.0}$ | $40.7_{\pm2.0}$ | $\mathbf{61.0_{\pm1.0}}$ | $75.5_{\pm0.8}$ |
| | | 10 | 9.29 | $58.8_{\pm1.6}$ | $58.8_{\pm1.5}$ | $60.4_{\pm1.1}$ | $35.9_{\pm5.7}$ | $48.3_{\pm3.3}$ | $48.7_{\pm2.9}$ | $\mathbf{68.7_{\pm1.7}}$ | |
| | int | 1 | 0.93 | $39.3_{\pm1.1}$ | $47.8_{\pm0.5}$ | $11.4_{\pm1.6}$ | $44.7_{\pm1.5}$ | $26.5_{\pm2.3}$ | $46.7_{\pm1.9}$ | $\mathbf{63.3_{\pm1.9}}$ | |
| | | 5 | 4.64 | $52.8_{\pm0.8}$ | $56.2_{\pm1.3}$ | $51.2_{\pm0.5}$ | $53.7_{\pm1.7}$ | $26.6_{\pm2.4}$ | $48.0_{\pm2.1}$ | $\mathbf{70.3_{\pm2.5}}$ | $85.7_{\pm0.5}$ |
| | | 10 | 9.29 | $62.1_{\pm2.2}$ | $61.7_{\pm1.3}$ | $61.2_{\pm2.4}$ | $61.1_{\pm3.0}$ | $44.1_{\pm3.3}$ | $56.1_{\pm2.0}$ | $\mathbf{76.5_{\pm1.9}}$ | |

other datasets and settings, showing that prior methods fail to model the spatio-temporal nature of event data.

**Our PACE lifts every baseline.** Equipping prior DD with our **PACE** (NCFM+PACE) restores and *amplifies* their advantage across *all 18/18* settings in Table 1. Gains are especially pronounced on dynamic streams: on DVS-Gesture with int at IPC=10, NCFM climbs from $56.1\%$ to $\mathbf{76.5\%}$ (+20.4%); with bin at IPC=10, it rises from $48.7\%$ to $\mathbf{68.7\%}$ (+20.0%). Similar improvements appear on other datasets/budgets (*e.g.,* CIFAR10-DVS int, IPC=50: $34.0\% \to \mathbf{49.7\%}$, +15.7%; N-MNIST bin, IPC=1: $67.1\% \to \mathbf{84.4\%}$, +17.3%), confirming the effectiveness of our methods.

**Coreset selection vs. NCFM+PACE.** Across datasets and budgets, learning *synthetic* events outperforms selecting real ones. On DVS-Gesture with integer grids at the same IPC, the best coreset trails our distilled set by a large margin (*e.g.,* at IPC=10, best coreset reaches only $61.2\%$, while ours achieves $76.5\%$). The gap stems from a fundamental capability difference: coresets directly select from the available real examples, while synthetic sets are *learned* to reconstruct class prototypes and optimize for the downstream objective, which is crucial under tiny budgets.

**Where the gains concentrate.** Benefits are largest for (i) **more dynamic** datasets (DVS-Gesture $\gg$ CIFAR10-DVS $\gtrsim$ N-MNIST) and (ii) **low-moderate budgets** (IPC $\leq$ 10). These regimes hinge on *temporal shape* (onset/offset, rhythm, peak density). PACE densifies spikes via residual membrane potential and aligns amplitude/phase statistics (CF-in-feature, FFT-in-time), recovering precisely these shapes from very few sequences.

**Closeness to full-data performance.** With only IPC=10 on DVS-Gesture int (*Ratio*=9.29% *of the training set*), the distilled set reaches $\mathbf{76.5\%}$, about $\sim89\%$ of the full-data upper bound ($85.7\%$). A similar proximity holds on N-MNIST int, IPC=10 (*Ratio*=0.17%): $90.0\%$ vs. $99.3\%$ ($\sim91\%$). On CIFAR10-DVS int, IPC=50 (*Ratio*=5%), performance reaches $49.7\%$ vs. $62.9\%$ ($\sim79\%$).

**Binary vs. integer event grids.** At matched budgets, int *tends* to outperform bin, especially on dynamic datasets (DVS-Gesture: IPC=1 $63.3\%$ vs. $50.7\%$, IPC=10 $76.5\%$ vs. $68.7\%$; CIFAR10-DVS: IPC=10 $41.3\%$ vs. $31.3\%$). Integer multiplicity yields smoother pre-activation trajectories and stabler normalization after temporal/spatial aggregation, which reduces gradient variance and batch jitter during distillation/training. Binary grids are threshold-sensitive. Small misalignments

get amplified by SNN nonlinearities. Our PACE partially mitigates this for `bin` by densifying spikes with residual membrane potential, yet `int` remains intrinsically advantageous on highly dynamic streams. A mild exception appears on N-MNIST at IPC=10, where `bin` slightly surpasses `int` (91.8% vs. 90.0%), likely due to simpler temporal structure where multiplicity adds limited benefit.

## 5.2 How many time bins are needed?

**More $T$ helps until saturation.** On DVS-Gesture (Table 2), `bin` improves near-monotonically with $T$, from 41.1% at $T=2$ to 62.8% at $T=10$ (+21.7 pts). In contrast, `int` peaks at $T=6$ (65.7%) and changes only marginally thereafter (64.6% at $T=8$, 65.5% at $T=10$). The full-data upper bound similarly saturates around $T \geq 8$ for `int` (about 87%), indi-

Table 2: Ablation of the time steps $T$ on **DVS-Gesture** (Accuracy %). We choose $T \in \{2, 4, 6, 8, 10\}$. The "Full dataset" row reports bin/int performance as *bin/int*.

| Type | $T=2$ | $T=4$ | $T=6$ | $T=8$ | $T=10$ |
|---|---|---|---|---|---|
| Full dataset | 65.1/80.7 | 75.5/85.7 | 79.5/86.6 | 80.3/87.1 | 81.4/87.1 |
| bin | $41.1_{\pm 7.1}$ | $50.3_{\pm 1.9}$ | $57.9_{\pm 3.6}$ | $59.8_{\pm 3.4}$ | $\mathbf{62.8_{\pm 3.1}}$ |
| int | $60.2_{\pm 2.3}$ | $63.3_{\pm 1.9}$ | $\mathbf{65.7_{\pm 1.4}}$ | $64.6_{\pm 1.9}$ | $65.5_{\pm 1.1}$ |

cating diminishing returns once dominant temporal rhythms are captured.

**Why `int` saturates earlier.** Integer encodes timing and intensity per bin, which smooths pre-activations and stabilizes normalization under temporal and spatial aggregation. This lowers gradient variance and batch jitter, so moderate $T$ is sufficient. Binary grids lack amplitude cues and are threshold-sensitive, thus they benefit more from higher temporal resolution. PACE densifies spikes and partly compensates for `bin`, yet `int` remains intrinsically easier to optimize at moderate $T$.

## 5.3 Ablation study of PEQ-Q and ST-DSM in our PACE

**Effect of $N$ in PEQ-N.** We conduct experiments on DVS-Gesture as shown in Table 3. For `int`, accuracy improves from 51.7% ($N=2$) to 54.9% ($N=4$) and peaks at 63.3% with $N=8$, indicating that a moderate codebook preserves multiplicity without over-fragmenting counts. Larger $N$ brings no fur-

Table 3: Ablation of codebook size $N$ in **PEQ-N** on **DVS-Gesture** (Accuracy %).

| Type | $N=2$ | $N=4$ | $N=8$ | $N=16$ | $N=32$ |
|---|---|---|---|---|---|
| bin | $\mathbf{50.7_{\pm 2.1}}$ | $49.0_{\pm 3.7}$ | $49.1_{\pm 2.9}$ | $47.5_{\pm 4.9}$ | $46.9_{\pm 6.5}$ |
| int | $51.7_{\pm 2.5}$ | $54.9_{\pm 1.7}$ | $\mathbf{63.3_{\pm 1.9}}$ | $59.3_{\pm 2.1}$ | $59.9_{\pm 1.9}$ |

ther gain and tends to create sparse bins that weaken gradients and stability. For `bin`, performance declines as $N$ grows ($50.7 \to 49.0 \to 49.1 \to 47.5 \to 46.9$), consistent with sharper quantization boundaries interacting poorly with thresholded inputs and amplifying jitter. In practice, use $N \approx 8$ for `int` and keep $N$ small (2) for `bin`.

**Ablation of ST-DSM.** We conduct ablation of the ST-DSM's components on DVS-Gesture as shown in Table 4. DSR denotes the densified spike representation in Eq. (9). Disabling ST-SM removes the FFT alignment, i.e., Eq. (10) $\to$ Eq. (4). We can conclude: **(i):** DSR is the primary driver and stabilizer: for `int` it lifts accuracy from 46.7/48.0/52.7 to 56.5/60.3/63.7 (IPC 1/5/10), whereas **(ii):** ST-SM alone is smaller or volatile (46.9/50.8/55.3). **(iii):** Combining DSR+ST-SM (ST-DSM) is consistently best, `int`: 63.3/70.3/76.5; `bin`: 50.3/61.0/68.7, and also reduces variance (*e.g.,* `bin` IPC= 1, std 2.1 vs 5~7).

Table 4: Ablation of ST-DSM on **DVS-Gesture**.

| IPC | integer grid | | | binary grid | | |
|---|---|---|---|---|---|---|
| | DSR | ST-SM | Acc | DSR | ST-SM | Acc |
| 1 | ✗ | ✗ | $46.7_{\pm 1.9}$ | ✗ | ✗ | $40.8_{\pm 5.1}$ |
| | ✓ | ✗ | $56.5_{\pm 2.7}$ | ✓ | ✗ | $47.6_{\pm 7.0}$ |
| | ✗ | ✓ | $46.9_{\pm 7.4}$ | ✗ | ✓ | $47.7_{\pm 5.0}$ |
| | ✓ | ✓ | $\mathbf{63.3_{\pm 1.9}}$ | ✓ | ✓ | $\mathbf{50.3_{\pm 2.1}}$ |
| 5 | ✗ | ✗ | $48.0_{\pm 2.1}$ | ✗ | ✗ | $48.1_{\pm 2.1}$ |
| | ✓ | ✗ | $60.3_{\pm 3.1}$ | ✓ | ✗ | $55.1_{\pm 2.9}$ |
| | ✗ | ✓ | $50.8_{\pm 2.3}$ | ✗ | ✓ | $52.1_{\pm 2.0}$ |
| | ✓ | ✓ | $\mathbf{70.3_{\pm 2.5}}$ | ✓ | ✓ | $\mathbf{61.0_{\pm 1.0}}$ |
| 10 | ✗ | ✗ | $52.7_{\pm 1.7}$ | ✗ | ✗ | $54.7_{\pm 1.7}$ |
| | ✓ | ✗ | $63.7_{\pm 1.1}$ | ✓ | ✗ | $55.1_{\pm 2.9}$ |
| | ✗ | ✓ | $55.3_{\pm 2.8}$ | ✗ | ✓ | $54.6_{\pm 4.4}$ |
| | ✓ | ✓ | $\mathbf{76.5_{\pm 1.9}}$ | ✓ | ✓ | $\mathbf{68.7_{\pm 1.0}}$ |

## 5.4 Discussion

**Visualization analysis.** Fig. 2 compares real events (top) with distilled `bin` (middle) and `int` (bottom) for (a) DVS-Gesture right hand wave, (b) N-MNIST 0, and (c) CIFAR10-DVS airplane. Across all classes, the distilled `int` sequences show clearer contours and more coherent temporal evolution than `bin`. For the hand wave, alternating ON and OFF bands track the oscillatory motion.

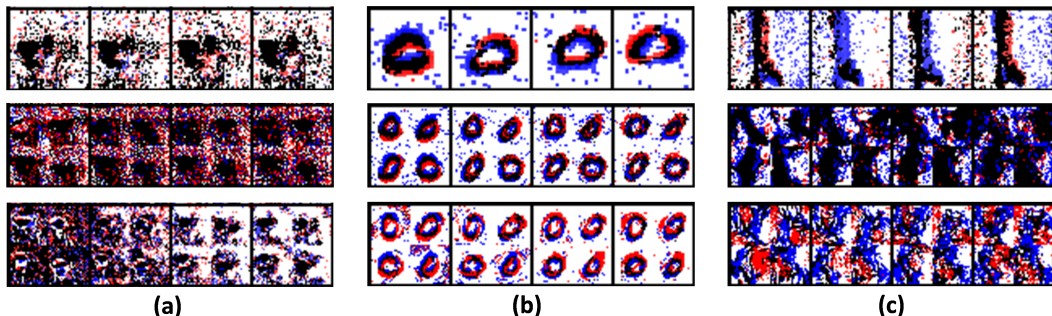

|  | (a) | (b) | (c) |

Figure 2: Visualization of Original real data (top), distilled binary (middle) and integer (bottom) event data for (a) DVS-Gesture "right hand wave", (b) N-MNIST "0", and (c) CIFAR10-DVS "airplane". Each subfigure shows representative voxelized event maps; red denotes positive (ON) events, blue denotes negative (OFF) events, and black marks pixels where both polarities occur within the same spatiotemporal bin.

For the digit 0, a smooth annulus appears with balanced polarities along the stroke. For the airplane, elongated motion edges emerge instead of salt and pepper noise. Mixed-polarity pixels (black) cluster on true edges for `int`, indicating plausible ON/OFF co-occurrence, while `bin` is sparser and more flicker-prone and often misses weak responses. Both distilled variants suppress background clutter relative to real data, and `int` preserves fine structure more faithfully, which matches its higher accuracy. Residual artifacts remain in the airplane case due to scene complexity, leaving room for refinement at very fine temporal scales. Overall, PACE yields compact and class-consistent spatiotemporal templates, with `int` providing stronger shape and rhythm fidelity than `bin`.

**Transferability.** We explore whether data distilled on VGGSNN can transfer well to distinct SNN backbones, as shown in Table 5. Compared to NCFM, our method improves SNN-ConvNet by **+17.1** pts (`bin`: 62.5 vs. 46.8) and **+13.4** pts (`int`: 68.2 vs. 54.8); SEW-ResNet by **+17.2** pts (`bin`: 29.6 vs. 12.4) and **+7.2** pts (`int`: 37.7 vs. 30.5); and the source VGGSNN by **+17.1** (`bin`) and **+16.6**

Table 5: Across-architecture generalization on **DVS-Gesture**. Synthetic data are distilled with VGGSNN and then used to train/evaluate other backbones.

| Method | Grid | VGGSNN | SNN-ConvNet | SEW-ResNet |
|---|---|---|---|---|
| NCFM | bin | $33.6_{\pm2.1}$ | $46.8_{\pm3.6}$ | $12.4_{\pm2.3}$ |
|  | int | $46.7_{\pm1.9}$ | $54.8_{\pm2.7}$ | $30.5_{\pm1.9}$ |
| NCFM+PACE | bin | $50.7_{\pm2.1}$ | $62.5_{\pm0.6}$ | $29.6_{\pm0.6}$ |
|  | int | $63.3_{\pm1.9}$ | $68.2_{\pm1.4}$ | $37.7_{\pm1.9}$ |

(`int`) points. The consistent gains across `bin`/`int` suggest PACE learns spatiotemporal statistics not tied to a specific architecture. Accuracy on SEW-ResNet is lower, hinting at a larger inductive gap for residual/gated dynamics. Extending transfer to residual families (including spiking ResNets) with residual-aware alignment is promising future work.

## 6 CONCLUSION

In this paper, we introduced **PACE**, an event native dataset distillation framework for rapid SNN training on event streams. To our knowledge, this is the first work that formulates and studies dataset distillation for SNNs with event data. First, PACE combines **ST-DSM**, which densifies spikes via residual membrane potential and aligns amplitude and phase with a characteristic function in feature space and a Fourier transform in time, with **PEQ-N**, a straight through probabilistic integer quantizer. Next, across DVS-Gesture, CIFAR10-DVS, and N-MNIST, PACE outperforms coreset selection and prior distillation in all settings, with the largest gains on dynamic streams and at low or moderate IPC. In particular, with IPC=10 on DVS-Gesture at a ratio of 9.29%, it reaches 76.5%, which is about 89% of full data performance. Moreover, ablations show that DSR is the main driver while ST-SM refines alignment, that integer grids usually beat binary grids, that a moderate PEQ-N codebook is best, and that more time bins help binary while integer saturates near $T=6$. In addition, distilled sets transfer to other SNN backbones, although residual families remain harder. Finally, this event native distillation reduces storage and wall clock time, enables minutes to converge training on neuromorphic streams, and provides a practical path to efficient edge deployment and reproducible distilled benchmarks.

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

## A   APPENDIX

### USE OF LARGE LANGUAGE MODELS (LLMS)

We used a large language model (LLM) solely as a writing assistant for *language editing*grammar correction, wording/fluency polishing, and minor rephrasing for clarityand for *retrieval and discovery* to surface potentially relevant related work and references. The LLM was *not* involved in research ideation, problem formulation, methodology or experiment design, coding, data analysis, result generation, or drawing conclusions. All candidate references returned by the LLM were screened and selected by the authors; all technical content and conclusions were authored and verified by the human authors, who take full responsibility for the paper. The LLM is not eligible for authorship. Further details of these uses are described in the paper.

### ETHICS STATEMENT

**Real-world significance. PACE** enables low latency and low power learning on event cameras by providing compact distilled datasets and minutes-to-converge training. This makes on-device SNN adaptation feasible on resource-limited platforms such as drones, AR headsets, wearables, and smart prosthetics. The approach can lower energy and cost during training and deployment, support privacy by keeping adaptation at the edge, and broaden access to neuromorphic perception in settings with tight memory and latency budgets.

