# OpenReview forum: "Minutes  to Converage: Dataset Distillation for Rapid SNN Training on Event Streams"
_ICLR.cc/2026/Conference — ICLR 2026 Conference Withdrawn Submission_

### Official Review · Reviewer_VDS1 · 2025-10-25

**Soundness:** 1
**Presentation:** 3
**Contribution:** 2
**Rating:** 2
**Confidence:** 4

**Summary:**

This study introduces PACE, the dataset distillation framework for SNNs. To achieve this, the ST-DSM module injects membrane potential residuals to obtain spike features. In addition, the PEQ-N module quantizes floating-point synthetic data into integer event frames using differentiable optimization. The experiments on DVS-Gesture, CIFAR10-DVS, and N-MNIST can achieve near-full-data accuracy with good compression ratios.

**Strengths:**

1. The paper is easy to understand and follow.
2. The extension of dataset distillation to the SNN scenario has certain inspirations for the neuromorphic community.

**Weaknesses:**

1. The proposed method highly relies on the gradient information of SNN. However, there are various gradient-free and brain-inspired training methods for SNNs. The proposed method seems like it will fail in such a scenario.
2. The study only explores the case that the teacher and student have the same architecture. If the teacher and student architecture are different, will the proposed method still work?
3. The static dataset can be considered as the “event-based” dataset by repeating T times. Can the proposed method handle such a dataset like ImageNet to speed up the training process? As we all know, training SNN on ImageNet is very time-consuming.
4. The paper only focuses on the CNN-based model. The spiking Transformer model should be selected as the teacher and student model.
5. Authors argue that the proposed method opens a path to efficient on-device learning but do not provide any experiment to justify it.
6. The DVS datasets used in this study are relatively small. To further verify the effectiveness of the proposed method, the large-scale DVS datasets must be considered.
7. Authors should also consider the real-world application cases and tasks of DVS devices.

**Questions:**

Please check the Weaknesses part.

---

### Official Review · Reviewer_E9uk · 2025-10-29

**Soundness:** 2
**Presentation:** 2
**Contribution:** 1
**Rating:** 2
**Confidence:** 4

**Summary:**

This paper proposes a data distillation method for efficiently learning weights in SNNs. The defining feature of the proposed method is to address not only spatial but also temporal features of the full event data. This was accomplished by ST-DSM applicable to SNNs with binary (and thus discrete) feature maps. Further, the authors proposed PEQ-N that forces the synthetic dataset samples to output integers (spike count within a given timestep). The proposed method outperforms a few previous methods in terms of classification accuracy for the same number of samples on CIFAR10-DVS, N-MNIST, and DVS-Gesture.

**Strengths:**

$\textbf{Performance compared with previous methods.}$ Given the consideration of spatio-temporal aspects of event data samples, the dataset distillation by the proposed method outperforms several previous methods in terms of classification accuracy for the same number of samples.

**Weaknesses:**

$\textbf{Limited impact of the proposed work:}$ In spite of the performance improvements compared to several previous works, the dataset distillation method still results in accuracy far behind the that from full dataset samples. Further, the complexity in this method is likely very high particularly for the phase-similarity measure (which is the square of number of timesteps).

$\textbf{Uncertainty in scalability:}$ The authors may argue that learning efficiency is achieved at the cost of loss in classification accuracy. However, the datasets under test in this paper are lightweight so that the efficiency gain should not compensate for the large loss in performance in accuracy. To address learning efficiency, the authors should address real-world event datasets of large complexity unlike the lightweight datasets.

$\textbf{Complexity in the proposed method:}$ Classification accuracy was taken as the only metric to compare this work with the previous works. The authors should report the complexity in the proposed method compared with the previous ones.

**Questions:**

What are the complexity of this method and any advantages over the previous methods?

---

### Official Review · Reviewer_sZA1 · 2025-11-01

**Soundness:** 2
**Presentation:** 2
**Contribution:** 1
**Rating:** 4
**Confidence:** 4

**Summary:**

This paper introduces PACE (Phase-Aligned Condensation for Events), a dataset distillation framework specifically designed for spiking neural networks (SNNs) operating on event-based data such as DVS streams. The method integrates ST-DSM (Spatial-Temporal Densified Spike Matching) and PEQ-N (Probabilistic Event Quantizer). The authors demonstrate that PACE can significantly improve dataset distillation performance across DVS-Gesture, CIFAR10-DVS, and N-MNIST datasets.

**Strengths:**

- The authors report mean ± standard deviation over multiple seeds and model initializations, demonstrating strong reproducibility and stability.
- The paper carefully separates the effects of PEQ-N and ST-DSM, showing each component’s contribution. Cross-backbone experiments demonstrate good generalization to unseen architectures.

**Weaknesses:**

- **Unclear Motivation:** The introduction asserts that SNNs discretize event streams into few time bins and *“increases storage, lengthens sequences, and raises optimization cost”* (Line 45-47). No quantitative evidence is provided to support that binning increases storage or that dataset distillation meaningfully reduces it. The datasets used (CIFAR10-DVS, N-MNIST, DVS-Gesture) are relatively small, making the motivation for “reducing storage” less compelling. It is unclear why sacrificing >10 % accuracy on such small datasets is beneficial.
- **Overclaimed Contributions:** The paper states that PACE *“demonstrates large reductions in storage and training time while maintaining high accuracy”* (Line 76-77). However, no explicit experiments quantify storage, training time, or wall-clock improvements. Moreover, a more than 10% accuracy drop reported on DVS-CIFAR10 seems to contradict the “high accuracy” claim.
- **Weak Baselines:** The real-data baselines are lower than existing state-of-the-art SNN results, such as Deng et al., *ICLR 2022*.

[Deng, et al. Temporal Efficient Training of Spiking Neural Network via Gradient Re-weighting. ICLR 2022.]

**Questions:**

1. Line 146: What does “gradient-matching DC” refer to? There are too many abbreviations introduced without clear definitions, or their explanations appear much later in the text, which makes the paper difficult to follow.
2. Line 147: The phrase *“and even more so for SNNs”* implies gradient matching is especially costly for SNNs. Can the authors provide evidence or cite benchmarks comparing gradient-matching cost between ANNs and SNNs?
3. How does dataset distillation for SNNs differ from that for conventional ANNs? Could the proposed PACE framework also be applied to ANNs? Moreover, what factors cause existing ANN-based dataset distillation methods to exhibit such a significant drop in accuracy when applied to event-based SNN tasks?

---

### Official Review · Reviewer_jrA1 · 2025-11-02

**Soundness:** 3
**Presentation:** 3
**Contribution:** 3
**Rating:** 6
**Confidence:** 4

**Summary:**

This paper presents PACE, a dataset-distillation framework tailored for event streams and SNNs that preserves causal spatiotemporal structure, unlike image-centric approaches. The method introduces ST-DSM for spike densification and phase-aligned matching via characteristic-function and DFT-based temporal alignment, and PEQ-N for differentiable integer event quantization. Using a time-expanded condensation loss with frozen teacher features, PACE achieves up to 89% of full-data accuracy with ~9% data and yields +20% gains over strong baselines on DVS-Gestures, CIFAR10-DVS, and N-MNIST, demonstrating effective and efficient event-data condensation.

**Strengths:**

1. First method to address dataset distillation for event streams and SNNs, with a phase-aligned condensation strategy tailored to event data.

2. Technically sound design combining spike densification and straight-through event quantization, consistent with neuromorphic computation characteristics.

3. Well-structured objective using frozen teacher features, helping maintain temporal causality and training stability.

4. Strong empirical results across multiple event datasets and IPC settings, showing consistent improvements and backbone generalization.

**Weaknesses:**

1. The paper centers on SNNs, but the motivation for excluding other temporal models (e.g., event Transformers, recurrent vision models) could be better justified. Clarifying whether PACE is fundamentally tied to SNN dynamics or could extend to other event-processing architectures would strengthen positioning.

2. The characteristic-function projection and phase-matching pipeline are mathematically dense. While theoretically sound, some readers may struggle without additional visual intuition or toy illustrations.

3. All baselines are image- or RGB-based distillation methods. It would improve fairness and clarity to include or discuss: Event-stream distillation/replay strategies (if any exist); Video distillation/distillation-in-time methods; Recent Transformer-based dataset condensation techniques (where applicable)

4. The title emphasizes "minutes-to-converge", but no wall-clock runtime or computational-cost table is included. Providing explicit speedups (vs. full training and vs. baselines) is crucial.

5. Potential ambiguity in the use of “bin”, sometimes referring to time bins, sometimes implying binary. Disambiguation would improve readability.

**Questions:**

1. Could PACE be extended to event Transformer architectures or time-continuous MLPs? Which components rely inherently on membrane-potential dynamics?

2. Can the authors provide a simple diagram or intuitive example illustrating how phase alignment improves synthetic event quality beyond feature-space matching?

3. Please report: Wall-clock training time (vs. full training + baselines); Energy/computation footprint (GPU hours / neuromorphic hardware cycles if relevant); Memory footprint of distilled synstreams vs. raw DVS data

4. Are there event-based replay/condensation works or video distillation baselines that could be included or discussed to situate PACE in a broader context?

5. How does PACE perform on: Higher-resolution real-world DVS datasets (e.g., DSEC)? More complex tasks (e.g., event optical flow, gesture recognition beyond classification)?

---

### Note · Authors · 2025-11-15

I have read and agree with the venue's withdrawal policy on behalf of myself and my co-authors.